# A Survey of Telemedicine Use by Doctors in District Hospitals in KwaZulu-Natal, South Africa

**DOI:** 10.3390/ijerph192013029

**Published:** 2022-10-11

**Authors:** Christopher Morris, Richard E. Scott, Maurice Mars

**Affiliations:** 1Department of TeleHealth, School of Nursing & Public Health, College of Health Sciences, University of KwaZulu-Natal, Durban 4041, South Africa; 2Department of Community Health Sciences, Cumming School of Medicine, University of Calgary, Calgary, AB T2N 4N1, Canada; 3Department of Digital Health Systems, College of Nursing and Health Sciences, Flinders University, Adelaide, SA 5042, Australia

**Keywords:** telemedicine, instant messaging, WhatsApp, legal, regulatory, ethical practice, guidelines

## Abstract

There is anecdotal evidence of informal telemedicine activity in KwaZulu-Natal (KZ-N), South Africa. Aim: To determine the current extent of telemedicine in district hospitals in KZ-N; the range of clinical activities and technologies used; additional services needed; current knowledge and practice regarding legal, ethical, and regulatory issues; and the need to formalise telemedicine activities. Method: A cross-sectional survey of telemedicine use by 143 doctors working at 22 District hospitals in KZ-N. Results: Most doctors (96%) participated in some form of telemedicine across a spectrum of disciplines, but more than half did not consider their activities to constitute telemedicine. To meet their needs, doctors have started their own informal services with colleagues, using mostly instant messaging and chat groups (WhatsApp). Some doctors indicated the need to formalise these services and establish additional services. Few doctors were aware of the national telemedicine guidelines and the required written informed consent for telemedicine was seldom obtained. This could have serious legal, regulatory, and ethical implications. Conclusions: Practical clinical and technical guidelines and standard operating procedures need to be developed with the active participation of the clinical workforce. These should encourage innovation and greater use of telemedicine, including the use of instant messaging apps.

## 1. Introduction

South Africa was an early adopter of telemedicine in the late 1990s [1]. Phase one of the National Telemedicine System was launched by the National Department of Health in 1999 at 28 sites in six of the nine provinces [1]. Services trialled were video-conferenced antenatal tele-ultrasonography, store and forward teleradiology, tele-ophthalmology, and telepathology [1]. All but the store and forward teleradiology failed within two years [1,2]. The reasons included limited and expensive bandwidth, inappropriate software, lack of trained technical support, lack of staff buy-in, and failure of the provincial departments of health to take ownership and budget for telemedicine [2]. This top-down approach failed, and the subsequent phases of the project were cancelled.

Other telemedicine activities continued at the Medical Research Council of South Africa, the Council for Scientific and Industrial Research, and several medical schools. These included store and forward teledermatology [3,4], pathology services [5], video-conferenced teledermatology, and postgraduate and continuing medical education [6,7]. The Moorfields Eye Hospital in England also provided international store and forward tele-ophthalmology to a hospital in KwaZulu-Natal (KZ-N) [8].

While the South African eHealth Strategy of 2012–2016 [9] describes telemedicine as an enabling tool to bridge the gap between rural healthcare and specialists, it also acknowledges the limited success of its past programme. Despite the growing use and ubiquity of mobile devices, the only mention of mHealth in the strategy was an unspecified “system” developed for the 2010 Soccer World Cup and a system to collect data for HIV counselling and testing. In contrast, the mHealth Strategy of 2015–2019 [10] (aligned with the eHealth strategy of 2012–2016) highlights the use of mHealth as an integral part of the delivery of healthcare and identifies the need for a security policy on the care and loss of mobile phones.

More recently, the National Digital Health Strategy for South Africa, 2019–2024 [11], states that telemedicine, including mHealth, should be used as a means for improving access to rural healthcare. Despite all these strategy documents produced over the last decade, few formal telemedicine activities are currently run by the KZ-N Department of Health (DoH).

The growing use of mobile phones, the evolution of smartphones, mobile phone applications (apps), and new forms of communication have facilitated rapid, relatively simple, and low-cost communication between clinicians [12]. WhatsApp use has been reported in several clinical services around the world, many of which began informally [13,14,15,16] and have grown further during the COVID-19 pandemic [17]. In 2013, anecdotal evidence suggested that doctors in rural hospitals in KZ-N were using WhatsApp for teledermatology and burns referrals [7]. Neither the teledermatology nor the burn telemedicine services were planned, there were no budgets, no training had been required, and the KZ-N DoH had no knowledge of their existence. In the absence of any formal telemedicine service, doctors had found a way to deliver a teledermatology and teleburns service [7,18]. There are further anecdotal reports of informal telemedicine activity within and between public hospitals in the province in several disciplines.

South Africa has a two-tiered, and highly unequal, healthcare system with 82% of the population dependent on state-funded public healthcare [19]. The private sector is funded through the contributions of individuals to private insurance or medical aid schemes and out-of-pocket payments. The state-sector hospital system in South Africa follows the traditional model of primary health clinics and district, regional, tertiary, and central hospitals with referrals following this upward path [7].

All district hospitals must offer the same defined spectrum of level 1 (generalist) services to in-patients and outpatients (ideally for referrals from a community health centre or clinic) and do not have a specialist staff [20]. Specialist staff at regional and tertiary hospitals are expected to participate in outreach programs to district hospitals where there are no specialists [7]. In the absence of onsite specialists, it is important to document current telemedicine activities, understand doctors’ needs for telemedicine, and address concerns relating to the legal, ethical, and regulatory use of telemedicine. The aim of this study was to gain an overview of the current status of telemedicine activities in district hospitals in KZ-N (describe the range of clinical activities and technologies used; identify additional services doctors would like to have; understand their current knowledge and practice relating to legal, ethical, and regulatory issues; and determine the need to formalise telemedicine activities).

## 2. Methods

To understand the current status of telemedicine activity in KZ-N, a cross-sectional survey was undertaken using a convenience sample of doctors from a randomised sample of district hospitals in KZ-N. In addition, the KZ-N Provincial DoH was asked to list the formal telemedicine services in KZ-N that they organise and for which they are responsible.

This study was undertaken in KwaZulu-Natal, the second most populous of the nine provinces in South Africa. It has a population of approximately 11.5 million people [21], 55% of whom live in rural areas [22] and nearly 88% of whom are dependent on public healthcare [23]. The province has 11 health districts and 37 district hospitals. District hospitals were chosen for the survey, as they do not have specialists on-site and refer patients to specialists at one of 13 regional or tertiary hospitals. The intent was to include two district hospitals from each of the eleven districts to gather a broad and representative sample. As one district had only one rural district hospital, an additional rural hospital was randomly selected from one of the other districts.

A cross-sectional survey was completed in late 2019 in 22 district hospitals in KZ-N, before the COVID-19 pandemic. The medical managers at the district hospitals were contacted telephonically, and the aim of the study was explained to them. They were asked to complete the questionnaire and request their doctors (typically about 8 per hospital) to complete the survey. The reluctance of some medical managers to participate necessitated the inclusion of three alternate sites. The survey was made available online using Google Forms or distributed and completed by hand at a continuing medical education event at each district hospital.

### Survey Instrument

A 43-item questionnaire covered five domains: respondents’ demographics, technology ownership and connectivity, use of telemedicine, and knowledge of legal and regulatory issues, including consent, data security measures, and guidelines for telemedicine practice (Appendix A). The questionnaire comprised twenty-nine multiple choice, two 5-point Likert scales, ten dichotomous (yes/no), and two open-ended questions. The questionnaire was pre-validated by five people involved in telemedicine and medical informatics, and questions were modified where necessary. Descriptive statistics were used to analyse the data.

This study was undertaken at the request of the eHealth Steering Committee of the KwaZulu-Natal Department of Health. Ethics approval was obtained from the University of KwaZulu-Natal (HSS/0185/018CA) and the KwaZulu-Natal Department of Health (HRKM265/18K2_201808_011) ethics committees. All respondents consented to participate.

## 3. Results

One hundred and forty-three responses were received from the 22 district hospitals. Based on the maximum staff complement of each hospital, there would have been a maximum of 266 returns. A total of 143 responses were received, representing a minimum response rate of 54%.

The respondents were predominantly male (79; 55%), of whom 40% were over the age of 50 years. The 64 female doctors were younger, with 81% under the age of 40 years. Of note was the limited response to certain questions related to legal and ethical issues. This posed issues in the analysis and, therefore, relevant denominators in the results are specified where appropriate.

### 3.1. Technology Access

All but one doctor owned a smartphone, the exception being a male over the age of 60 years who used a desktop computer with internet and chat for administrative purposes. In the hospitals, although most respondents (117/143; 82%) had access to a computer, only 93 (65%) had access to a computer with an internet connection. Nineteen doctors (13%) reported having wireless access within defined zones at eight hospitals. All the hospitals had mobile phone coverage, although a poor network signal was reported at seven sites. A number of open-ended comments were made regarding poor or no internet access or Wi-Fi. Seventy-nine doctors responded to a multiple-choice question regarding connectivity, and of these, all of them used their own data bundle on their mobile phones. In addition, one respondent also used a hospital-networked computer, and one noted the use of Wi-Fi paid for by doctors at their work.

### 3.2. Telemedicine

Some form of telemedicine activity was reported by most respondents (137; 96%) at all 22 district hospitals. The six doctors who did not report telemedicine practice all owned smartphones, and two indicated they would like to use telemedicine services. Many of the respondents did not perceive their use of instant messaging and chat groups, phone consultations, or even video conferencing as constituting telemedicine (77/137; 56%). Different modes of communication were used for telemedicine consultation, seeking a second opinion or diagnosis, giving management advice, and education (Table 1).

Instant messaging was used by almost all of the respondents (136/143; 95%) and was used daily or weekly by most respondents (79%) in a wide range of clinical services (Table 2). In addition, doctors were asked what additional telemedicine services they would like (Table 2).

Chat groups were used in all of the hospitals by 119 doctors (83%) for a range of activities. Most used chat groups for informing colleagues about the status of a patient (107; 90%), informal discussions about a clinical problem (107; 90%), seeking advice about a patient (98; 82%), giving advice to junior staff (85; 71%), and continuing medical education (60; 50%). In addition, chat groups were used for administrative purposes, including booking theatres and organising meetings (106; 90%). Mobile phone calls were the next most common mode of telemedicine communication used by the respondents (70/137; 51%), of whom 36 made mobile phone calls daily (51%), 21 weekly (30%), and 13 (19%) ad hoc.

Video conferencing had been used for clinical services by 20/143 respondents (14%); five doctors had used it weekly, seven on an ad hoc basis, and eight once a quarter (Table 2). At two sites, two doctors reported that the consulting paediatricians required them to send a video of infants in respiratory distress prior to their consideration of accepting transfers. 

When asked if they used telemedicine, 82/143 (57%) responded that they did not. As noted above, based on their responses to other questions, nearly all of them used instant messaging and or chat groups (76/82; 93%); one used only video conferencing, and 25/82 (30%) made mobile phone calls related to clinical management. Thus, many did not perceive that their use of instant messaging, chat groups, video conferencing, or voice calls constituted telemedicine. Of the 143 respondents, 75 gave one or more reasons for not using telemedicine (Table 3).

### 3.3. Telemedicine Satisfaction

Satisfaction with telemedicine activities was assessed on a 5-point Likert scale, where a score of 5 was the most satisfied. Eighty-seven doctors (61%) responded to the question, and the median score was four. Forty-five respondents (52%) reported satisfaction, fourteen (16%) were dissatisfied, and twenty-eight (32%) were unsure. Of the fifty-six (64%) who did not respond, fifty used instant messaging and or chat groups, and six did not use telemedicine.

### 3.4. Clinical Services Desired

Additional services that 134 doctors wanted but have had difficulty obtaining included acute medical care (89; 66%), wound care (80; 60%), psychiatry (65; 49%), maxillofacial surgery (51; 38%), and dentistry (32; 24%) (Table 2). Teleradiology was identified as a desirable service by 111 respondents (83%), and if a formal teleradiology service was to be made available 127/129 (98%), the respondents indicated they would use it, of whom about half (65/127; 51%) would refer a median of 2–5 sets of X-rays per week, while five other doctors would refer more than 20 sets of X-rays per week. If a formal instant messaging teledermatology service was available, nearly all of the respondents (132; 92%) felt that they would use it weekly (82; 57%), monthly (20; 15%), quarterly (1), or ad hoc (29; 20%). Of the 62 doctors not currently using instant messaging for teledermatology, 51 (84%) would like to do so.

### 3.5. Open-Ended Questions

The verbatim responses of the nine doctors who responded in writing to the open-ended questions are presented.

Four doctors reported their views on the benefits of telemedicine:


*“My usage mainly with Clinic sisters WhatsApping CTG [cardiotocograph] traces for comment.”*
(Dr 65)


*“Telemed can help us in cutting time spent by each person on daily bases. currently our patient spend three days just to access specialist clinic in PMB [Pietermaritzburg]. The shuttle leaves our hospital at 2 am and back at 7 pm. This is inconvenience for the sick patients. They often go for review by specialist, this could be done online/telemed if facilities were right.”*
(Dr 90)


*“I have used telemedicine in my previous job and it worked well for patients who didn’t have to travel to see a dermatologist.”*
(Dr 12)

Two doctors reported the need to formalise telemedicine:


*“There would be a great benefit in establishing formal telemedicine channels.”*
(Dr 23)


*“Need to formalise telemedicine.”*
(Dr 38)

The need for internet, wireless connectivity, and a data allowance was noted:


*“Healthcare facilities need to have WiFi so that telemedicine can function and HCPs [Healthcare professionals] need to be provided with a data allowance monthly.”*
(Dr 27)


*“We would need good Internet connectivity and support for our computers.”*
(Dr 58)

It was noted how WhatsApp was used to send a video:


*“We are required to send videos/pics to our referring institution whenever we are faced with a severely ill child that cannot be managed at our level (after a telephone discussion). This happens daily, sometimes two per day. Most patients require ventilation etc. Once videos/pics are sent and seen by regional then it is deleted from my phone.”*
(Dr 8)

With respect to the Health Professions Council of South Africa (HPCSA):


*“Telemedicine has great potential. HPCSA must come on board.”*
(Dr 139)

### 3.6. Guidelines

The respondents were asked if they were aware of the guidelines for telemedicine practice from any of the HPCSA, the South African Medical Association, the World Medical Association, the Medical Defence Union, or the National Department of Health. Nearly two-thirds of the respondents (93/143; 65%) did not answer the question. Overall, only 23/143 respondents (16%) were aware of the HPCSA ethical guidelines for telemedicine, 26/143 (18%) said they were aware of other guidelines attributed to the South African Medical Association (13), the National Department of Health (10), the Medical Defence Union (2), or the World Medical Association (1), and 14/143 (10%) were unaware of any guidelines. Six respondents were aware of two or more guidelines.

### 3.7. Consent

Seventy-nine of the one-hundred and thirty-seven respondents using telemedicine completed this question (58%). Of the 23 respondents who were aware of the HPCSA guidelines and the requirement for written consent, none routinely obtained written consent for the use of video conferencing or instant messaging. Three of these doctors obtained written consent on occasion when using video conferencing and two when using instant messaging. Ten of the twenty doctors using video conferencing (50%) and seven of the one-hundred and thirty-six using instant messaging (5%) obtained some form of consent. Consent was considered to be implied for both video conferencing and instant messaging by two each. Consent was mostly verbal or implied. Overall, 58 of the 137 doctors using telemedicine (42%) did not respond to the consent question (Table 4). Of the seventeen respondents who only used chat groups, just one obtained verbal consent to share patient information, and one felt that consent was implied. The remainder did not answer the consent question.

### 3.8. Mobile Phone Data Security

All but seven respondents (129/136; 95%) used one or more forms of mobile phone security. These were: the use of a password or PIN (111; 82%), fingerprint authentication (77; 57%), remote wiping (21; 15%), full device encryption (16; 12%), and a third-party lock screen app (5; 4%).

### 3.9. Formal Telemedicine Service Provided by the KZ-N DoH

The KZ-N DoH was contacted and asked what formal telemedicine services they provided. Only a limited teleradiology service between Nkosi Albert Luthuli Central Hospital and five regional hospitals was reported.

## 4. Discussion

There is widespread informal telemedicine activity in all of the district hospitals surveyed in KZ-N, despite there being no formal services organised by the KZ-N DoH at the district hospital level. To meet their needs in a number of specialities, the doctors have started their own informal services with colleagues, using mostly instant messaging and chat groups. Some doctors indicated the need to formalise these services and many to establish additional services. Although most doctors participated in some form of telemedicine, more than half did not consider their activities to be telemedicine. An implication of this is that these doctors would not see the need to comply with current guidelines and regulations. Only 23 doctors (16%) were aware of the General Guidelines for the Good Practice of Telemedicine from their regulatory authority—the HPCSA. As a result, written informed consent for telemedicine, an HPCSA requirement, was seldom obtained. This could have serious legal, ethical, and regulatory implications in the event of a clinical misadventure or a breach of patient confidentiality or privacy. A recent study in KZ-N reported a higher figure of awareness of the HPCSA guidelines (38.4%) [24], but this could be attributed to most responses being from doctors working within urban settings and in regional and tertiary hospitals.

Many respondents appear to have had a limited understanding of the meaning or concept of telemedicine as defined by the HPCSA (“the practice of medicine using electronic communications, information technology or other electronic means”) [25]. This would include all forms of electronic communication, ranging from a simple telephone call between healthcare providers to extremely complex applications involving remote-controlled surgery [26]. More than half of the respondents reported that they did not use telemedicine, citing lack of awareness and training, no technical support, and no availability of video conferencing equipment as reasons. However, nearly all of those who reported not using telemedicine actually used several modes of electronic communication, mostly instant messaging and chat groups, and mobile phone calls. This may be due to the perception that telemedicine is confined to video conferencing as in the failed National Telemedicine System, as only a few sites had video conferencing facilities and equipment. The low reported use of telemedicine in this study (39%) is similar to a recent study in KZ-N that reported telemedicine use at 20%; most of the responses were from doctors working within urban settings and reflected the larger hospitals in these settings [24].

Other than the KZ-N DoH teleradiology service, it is understood that there is only one other formal telemedicine service in KZ-N; the video conference-based teledermatology service run by the local medical school [27]. However, 20 doctors from five sites reported using video conferencing over a range of specialties. Some video conferencing may be attributed to video calls using smartphones, or the use of programmes such as Skype or Zoom, rather than using dedicated video conferencing equipment. In some instances, an acceptance of the transfer of infants in respiratory distress was dependent on the specialist first seeing a video of the infant.

Clinicians often have an urgent need to communicate with colleagues. Due to a limited telemedicine infrastructure in the hospitals (e.g., video conferencing equipment or workplace access to a personal computer with internet or wireless internet), they have resorted to using instant messaging and chat groups for telemedicine, at their own expense [28]. All but one doctor in this study owned a smartphone, and most doctors used instant messaging and chat groups for a range of clinical services. While there is an informal activity in a number of specialities, many respondents wanted additional services, particularly in acute medical care, wound care, plastic surgery, psychiatry, maxillofacial surgery, and dentistry. Most respondents said they would use a teleradiology service and nearly all an IM teledermatology service if they were available.

Nearly two-thirds of the respondents did not answer the question on awareness of telemedicine guidelines. Furthermore, just under half of the doctors using telemedicine did not respond to the consent question indicating either a lack of knowledge or disregard for HPCSA guidelines. It may also reflect concern that their response would show that they are not following the prescribed practice regarding consent and, thus, making themselves potentially liable for sanction if identified. Regardless, due to the apparent lack of awareness of the HPCSA’s General Ethical Guidelines for Good Practice in Telemedicine in South Africa (Booklet 10) [25], there are clear concerns about the intersection of reality and the requirements of the guidelines that need to be addressed [29].

A number of other privacy concerns are raised with the use of telemedicine [30,31]. When video conferencing is used, how does the patient know who is listening or watching, how many people are in the room at the receiving site, or that their identity may be revealed to such people, and they must consent to this? A further privacy issue arises with patient identification when using instant messaging. The HPCSA requires that all personal identification be removed from messages. How, then, does the specialist, or members of the chat group, confidently identify the patient in order to avoid confusion and possible clinical misadventure? Furthermore, if the patient’s identity is removed from the information, how can the doctor maintain a patient record, or how can the patient’s record be retrieved? Workarounds, such as giving the patient’s ward, bed number [32], and/or the consultant’s name, or identifying the patient by gender and birthdate, do not ensure anonymity [33] nor an unambiguous identification. A further privacy issue concerns the use of photographs; how is a de-identified photograph (in a message that maintains anonymity) going to be correctly assigned to the record that should be kept by the doctor from whom advice is being sought? The HPCSA does not offer guidance for this.

This study has clearly shown widespread informal telemedicine activity in KZ-N pre-COVID. Ironically, the use of telemedicine was still considered the exception rather than the rule shortly before the COVID-19 pandemic, even in the USA, where significant local and national policies and regulations have striven to curry interest [34]. This has changed globally with the onset and prolonged period of the pandemic, causing a significant shift towards telemedicine consultations [17,35,36]. As of May 2020, the use of telemedicine had increased in over 125 countries across the globe [37]. For example, it had accelerated the adoption of smartphone and digital technologies [38], instant messaging apps, such as WhatsApp and iMessage [39], and video conferencing apps, such as Skype and Zoom, for clinical communication and remote consultation [40]. Healthcare providers implemented these delivery models to ensure accessibility and continuity of patient care [41].

Early in the pandemic, technology adoption within Africa reportedly addressed three categories of action (disease prevention, disease surveillance, and supply management) in order to manage or monitor patients and provide accurate COVID-19 information [42]. These efforts were both sovereign (national, sub-national, and non-governmental organisations), as well as international agency-initiated (Africa Centres for Disease Control and Prevention). A surprisingly wide variety of technologies were used, with relatively few associated with instant messaging. As the pandemic rolls into mass vaccination programmes, mobile technology solutions have been identified as an important option [43]. Within South Africa specifically, widespread use of technology has been made to provide diverse COVID-19-related healthcare services, which included a WhatsApp helpline, plus other approaches (SMS-based solutions, mobile health applications, telehealth (telemedicine), artificial intelligence, chatbots, and robotics) [44]. Many of these approaches were noted to have challenges and limitations, extending from fundamental regulatory, cultural, and interoperability issues, through cost and digital divide issues, to security and trust issues. Within the digital health-COVID-19 literature, the desire to maintain and maximise the use of technology solutions into the future is a common theme [42,44,45,46].

With the pandemic stimulating the need to promote the use of telemedicine, changes were made to national telemedicine policies around the globe. For example, the USA temporarily relaxed various telemedicine restrictions and provided additional telemedicine funding [47,48], and in Korea, patient–doctor telemedicine was temporarily applied to the entire population [49]. In the UK, the National Health Service relaxed their ban on doctors bringing and using their own devices and advised clinicians to draw on their smartphones and apps (e.g., Skype, iMessage, and Zoom) for telemedicine activities [40]. Even in South Africa, the HPCSA amended its telemedicine guidelines during COVID-19 to permit a telehealth consultation without a prior practitioner–patient relationship [50].

Within many countries [13,14,15,32], and as shown in this study, the need for real-time multimedia communication between doctors has resulted in the widespread use of instant messaging, circumventing the need for expensive video conferencing equipment or computer networks. It has been noted that “the advent of “spontaneous services” flaunts the traditional approach yet may be most likely to lead to successful integration and sustained application, particularly in the less regulated developing world setting” [7].

South Africa could continue to emulate the global shift towards telemedicine and supportive health policies and capitalise on new innovations to achieve its eHealth goals and enhance the quality of and access to medical care. However, the current spontaneous use of instant messaging in a wide range of clinical and administrative settings is not without fault and needs support and viable guidance in order to bridge the gap between rural healthcare and specialists.

### Limitations of the Study

The results reflect only the practices and opinions of doctors from 22 of the 37 district hospitals and are further limited to those who chose to answer the questionnaire. Also, as noted, some respondents appear to have avoided responding to some potentially contentious questions or to do so through a lack of understanding that their actions constituted telemedicine.

## 5. Conclusions

The informal use of instant messaging by clinicians in KZ-N district hospitals is widespread and shows that doctors have, in the absence of any formal service, found different ways of meeting their needs. However, the benefits of instant messaging must be balanced with the need to protect patient confidentiality and privacy, comply with data security laws, and maintain patient records. The HPCSA guidelines are not helpful for instant messaging, having largely been formulated prior to the development and widespread use of instant messaging in healthcare. Practical guidelines need to be developed with the active participation of the clinical workforce that encourages innovation and the enhanced use of technology, including instant messaging apps such as WhatsApp.

## Figures and Tables

**Table 1 ijerph-19-13029-t001:** Use and mode of telemedicine services (*n* = 137).

Mode	Consultation	Second Opinion	Seeking a Diagnosis	Giving Management Advice	Education
Instant messaging and chat groups	136	133	134	135	71
Mobile phone calls	21	36	37	33	10
Email	2	3	3	3	3
Video conferencing	19	16	17	20	3
Social media	2	2	5	4	1
Websites	1	4	8	7	8

**Table 2 ijerph-19-13029-t002:** Range of specialities for which telemedicine is used, telemedicine mode used, and additional telemedicine services doctors would like to have.

	Doctors (*n* = 137)	Doctors (*n* = 20)	Doctors (*n* = 136)	Doctors (*n* = 134)
Specialty	Currently Use TM	Currently Use VC	Currently Use IM	Services Requested
Dermatology	87	16	87	116
Paediatrics	78	7	78	115
Orthopaedics	70	9	70	105
Burns	60	9	60	100
Radiology	29	6	29	111
Cardiology	26	3	26	75
Obstetrics	21	5	21	80
ENT	17	1	16	22
Urology	11	1	11	66
High care	8	0	8	47
Psychiatry	6	1	6	65
Internal medicine	4	0	4	0
Dentistry	2	0	2	32
Wound care	1	0	1	80
Maxillofacial	1	0	1	51
Gynaecology	1	0	1	0
Ophthalmology	1	0	1	0
Family medicine	1	1	1	0
Plastic surgery	0	0	0	46
Acute medical care	0	0	0	89

TM = Telemedicine, VC = Video Conferencing, IM = Instant Messaging.

**Table 3 ijerph-19-13029-t003:** Reasons given for not using telemedicine (*n* = 75).

Reason	*n =*	Reason	*n =*
No technology support	31	Do not know what to do	11
No training in telemedicine	24	Did not know about it	11
Video conferencing not available	21	Absence of telemedicine guidelines	6
No services available	21	Legal and ethical concerns	2
No equipment	17	Have other concerns	1

**Table 4 ijerph-19-13029-t004:** Form of consent obtained by doctors (*n* = 79) for electronic communication based on their awareness of the HPCSA requirements.

	Video Conferencing		Instant Messaging	
	HPCSA Consent Requirements			
Type of Consent	Aware	Unaware	Aware	Unaware
Written only	0	3	0	2
Written or verbal	1	2	1	3
Written or verbal or implied	2	5	1	2
Verbal only	7	31	5	31
Verbal or implied	0	14	0	15
Implied only	2	7	2	9
None	1	5	1	5

## Data Availability

The data presented in this study are not publicly available due to privacy restrictions.

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
