# Peer review of "A Survey of Telemedicine Use by Doctors in District Hospitals in KwaZulu-Natal, South Africa"

_ijerph, 2022, doi:10.3390/ijerph192013029_

Round 1
Reviewer 1 Report
The aim of this paper was to assess for the prevalence of telemedicine usage in South Africa and to assess for prevalence of regulatory and ethical compliance.
It is applaudable that the research team would want to capture comprehensive data on telemedicine usage, which is gaining popularity around the world.
To that end however, I find the study design to have too many outcomes for a study size of 143 subjects. Due to the multiple outcomes being assessed, it was difficult to understand the overall study aims and the results reported. The lack of in-depth analysis (likely due to multiple outcomes) was understandable but not advised. Additionally for various reasons, it seemed that not every subject completed all the portions of questionnaire reducing sample size power.
The inclusion of textual responses from 9 subjects to the open-ended questions did not further contribute to the overall paper. It was difficult to assess whether the responses from 9 subjects were representative of the overall cohort given lack of data.
While the questions asked in the survey were broad, they did not contribute to the overall understanding of telemedicine in South Africa due to a lack of depth.
Due to the multiple outcomes being studied, ultimately it was difficult to find the main purpose of the paper which reduced reader's clarity and insight that could have been gained from the study.
Author Response
Please see attachment for Reviewers 1, 2 & 3.

Reviewer 2 Report
Overall, this is an informative analysis of telemedicine in one South African province. There are several edits and clarifications suggested below.
Methods:
The details of how the sample was derived are incomplete and the aspects included are not clear. How many surveys were distributed? What percent of people surveyed responded?
A figure that shows # sent out, response rate, any lost, any incomplete, etc. (a flow diagram) would strengthen the paper greatly.
This expression of numerator/denominator in general would strengthen the results. The discussion of how many districts, hospitals, etc. is very confusing and should be rewritten for clarity. Can more detail be provided on the hospitals that were included that were not district hospitals? Can the reader assume that all of the hospitals chosen were approximately the same? Is there statistical info on these hospitals that would reassure the reader that they are approximately the same?
If this journal allows, it would be helpful to include the survey as an appendix
There is no mention of IRB review. Some journals won’t publish a paper that does not reference work that was approved by an IRB.
Table 1 is laid out in a way that makes it unclear. Once again, what is the denominator, e.g., 136 respondents used instant messaging for consultation – I think that is what is meant – but out of how many respondents? Table 2 is more clear in this way.
The value of the quotes from open-ended questions is not clear. It may make sense to put them in a table as opposed to the text.
Cut the discussion by 50%
Author Response
Please see the attachment Reviewers 1, 2, & 3.

Reviewer 3 Report
The authors describe the extend of telemedicine use among clinicians in district hospitals in South Africa with an intend to understand their preferences, knowledge ethical/legal concerns that will enable implementation of further services within the system.
This reviewer has only 1 major concern. The authors are talking about telemedicine through out the manuscript which in reality refers to delivery of clinical services through the use of technology and this involves clinicians and patients. However, from the results it looks like they were talking about communication within clinicians rather than between patient and clinician. This looks like a descriptive study of the use of social media ( or a technology similar to tiger text) among clinicians for healthcare related matters. This is entirely different from Telemedicine and if this is the case, the title and abstract are giving a wrong impression. It is unlikely that the clinicians communicate with patients through WhatsApp as indicated in the article which itself is a major ethical issue. The authors need to use the correct terminology as Telehealth, telemedicine, m-health, e-health all have different meanings although they overlap slightly and revise the manuscript accordingly. The social media use by clinicians is also a very important topic that is not explored well and it will pave way for better guidelines and innovative platforms for the use of these technology in healthcare.
Author Response
Please see the attachment for Reviewers 1,2, & 3.

Round 2
Reviewer 3 Report
None
Author Response
We thank Reviewer 3 and note that no further comments or suggestions were made so no response given. We have therefore accepted our changes to the draft attached.
Please confirm.
